# Engineering induction of singular neural rosette emergence within hPSC-derived tissues

Gavin T Knight[1,2†], Brady F Lundin[1,2], Nisha Iyer[2], Lydia MT Ashton[3], William A Sethares[4], Rebecca M Willett[2,4], Randolph Scott Ashton[1,2]*

[1]Department of Biomedical Engineering, University of Wisconsin-Madison, Madison, United States; [2]Wisconsin Institute for Discovery, University of Wisconsin-Madison, Madison, United States; [3]Department of Consumer Science, University of Wisconsin-Madison, Madison, United States; [4]Department of Electrical and Computer Engineering, University of Wisconsin-Madison, Madison, United States

*For correspondence:
rashton2@wisc.edu

[†]These authors contributed equally to this work

Competing interests: The authors declare that no competing interests exist.

**Abstract** Human pluripotent stem cell (hPSC)-derived neural organoids display unprecedented emergent properties. Yet in contrast to the singular neuroepithelial tube from which the entire central nervous system (CNS) develops in vivo, current organoid protocols yield tissues with multiple neuroepithelial units, a.k.a. neural rosettes, each acting as independent morphogenesis centers and thereby confounding coordinated, reproducible tissue development. Here, we discover that controlling initial tissue morphology can effectively (>80%) induce single neural rosette emergence within hPSC-derived forebrain and spinal tissues. Notably, the optimal tissue morphology for observing singular rosette emergence was distinct for forebrain versus spinal tissues due to previously unknown differences in ROCK-mediated cell contractility. Following release of geometric confinement, the tissues displayed radial outgrowth with maintenance of a singular neuroepithelium and peripheral neuronal differentiation. Thus, we have identified neural tissue morphology as a critical biophysical parameter for controlling in vitro neural tissue morphogenesis furthering advancement towards biomanufacture of CNS tissues with biomimetic anatomy and physiology.
DOI: https://doi.org/10.7554/eLife.37549.001

## Introduction

The derivation of organoids from human pluripotent stem cells (hPSCs) has redefined the possibilities of in vitro tissue engineering. Human PSCs have traditionally been characterized by their ability to spontaneously recapitulate facets of developmental tissue morphogenesis upon ectopic implantation in vivo, that is the teratoma assay (*Thomson et al., 1998*). However, more recent discovery of their ability to also self-organize emergence of organotypic tissues in vitro, for example cortical (*Eiraku et al., 2008*), retinal (*Nakano et al., 2012*), cerebral (*Lancaster et al., 2013*) and intestinal organoids (*Watson et al., 2014*), has spurred the creation of novel developmental and disease models with unprecedented biomimicry of microscale cytoarchitectures and cell phenotype diversity (*Lancaster and Knoblich, 2014*; *Sasai, 2013*). The innate emergent properties of hPSCs could serve as the basis for advanced biomanufacture of functional tissue and organ models or even transplants (*Watson et al., 2014*; *Workman et al., 2017*). Yet, it remains a challenge to engineer reproducible hPSC morphogenesis in vitro and thereby enable controlled emergence of organotypic tissues with standardized cytoarchitecture (*Marti-Figueroa and Ashton, 2017*; *Lancaster et al., 2017*).

Human PSC-derived neural organoids commence with a pseudoneurulation event generating pseudostratified neuroepithelial cells (NECs). In this dynamic process, NECs polarize adherens,

for example N-cadherin, and tight junction proteins towards an apical lumen while depositing extra-cellular matrix (ECM) proteins at their basal surface (*Eiraku et al., 2008*; *Lancaster et al., 2013*; *Gilbert, 2006*). In vivo, neurulation results in emergence of a singular neuroepithelial tube that spans the entire rostrocaudal (R/C) axis of the embryo's dorsal plane and serves as the primordium of all CNS tissues. This singular polarized neuroepithelium serves as a critical morphogenesis center for organizing establishment of lamellar tissue cytoarchitectures during subsequent stages of development. For example, it localizes waves of mitotic NEC proliferation at the tube's apical surface while daughter cells migrate radially towards the basal surface to complete differentiation, functional maturation, and expand the CNS parenchyma (*Gilbert, 2006*; *Lui et al., 2011*). Disruption of the neuroepithelial tube's N-cadherin polarized cytoarchitecture eliminates the sole neurogenic source of embryonic CNS development (*Zhang et al., 2010a*). Alternatively, the presence of multiple neural tubes during development causes congenital abnormalities such as dipolmyelia and diastematomyelia (*Testoni et al., 2010*; *Cogliatti, 1986*) or anencephaly as observed in diprosopic parapagus twins (*Spencer, 2000*). Therefore, formation of a singular neuroepithelium within hPSC-derived organotypic tissues is an important cytoarchitectural feature for generating biomimetic CNS tissue models (*Lee et al., 2017*).

In current 2- and 3-D hPSC-derived cultures, neurulation-like events occur spontaneously yielding uncontrolled emergence of numerous polarized neuroepithelial tissues, a.k.a. neural rosettes (*Lancaster et al., 2013*; *Zhang et al., 2001*). Integration of engineering techniques such as stirred-tank bioreactor culture during 3-D neural organoid derivation can induce the formation of larger and more contiguous neuroepithelial tissues (*Lancaster et al., 2013*; *Renner et al., 2017*). Likewise, derivation of neural organoids around filamentous biomaterial scaffolds was shown to induce elongated neuroepithelial tissues within the 3-D organoid (*Lancaster et al., 2017*). Yet, the persistent presence of multiple neural rosettes of indiscriminate shapes and sizes within a single organotypic tissue inevitably confounds subsequent morphogenesis events and potentially limits tissue maturation. This also produces significant variability in the cytoarchitecture both within and between organoids of a single production batch. Interestingly, singular neural rosette emergence was routinely observed upon derivation of 3-D neuroepithlelial cysts, which have considerably smaller dimensions than hPSC-derived organoids and are generated from single cell mouse embryonic stem cell (mESC) suspensions (*Meinhardt et al., 2014*; *Ranga et al., 2014*). Manual isolation of singular neural rosettes from culture remains the only reliable in vitro method for generating such critical biomimetic cytoarchitecture within hPSC-derived organotypic CNS tissues (*Lee et al., 2017*; *Shi et al., 2012*).

Geometric confinement of hPSC tissues on 2-D micropatterned substrates was recently shown to induce self-organized embryonic patterning, that is mimicking gastrulation, in a morphology dependent manner (*Warmflash et al., 2014*; *Etoc et al., 2016*). Based on this and previously discussed mESC-derived neuroepithlelial cysts studies, we investigated whether engineering the morphology of neurally differentiating hPSC tissues could similarly regulate neural rosette emergence. Using a custom image analysis algorithm and machine learning classifier, screens of hPSC-derived neuroepithelial tissues of various micropatterned morphologies revealed an indirect correlation between the tissue's surface area and the propensity for singular neural rosette emergence. Circular micropatterns of 200 – 250 µm diameter (0.031 – 0.049 mm$^2$) were observed to most effectively induce singular neural rosette emergence within forebrain neuroepithelial tissues reaching levels of 80 – 85% efficiency when initially seeded as hPSCs. These studies were further extended to develop an analogous protocol for neuroepithelial tissues of a ventral spinal cord regional phenotype. This revealed that spinal neuroepithelial tissues, unlike their forebrain counterparts, preferentially formed singular neural rosettes (73.5%) on circular micropatterns of 150 µm diameter (0.018 mm$^2$), indicating novel biophysical differences between NECs of varying regional phenotypes. During the rosette emergence period, the proliferative neural tissues morphed from a 2-D monolayer to a 3-D, multi-layered, disk morphology of the prescribed micropatterned diameter. Upon release from micropattern confinement, the tissue slices expanded radially while maintaining a singular neuroepithelial rosette as the morphogenesis center, depositing a basal lamina, and displaying neuronal differentiation at the tissues' periphery. Thus, we have developed a culture platform for engineering controlled neural rosette emergence within hPSC-derived organotypic CNS tissues. Reproducible induction of this nascent cytoarchitecture is a pre-requisite for advanced biomanufacture of human CNS organoids with consistent, biomimetic anatomy (*Marti-Figueroa and Ashton, 2017*).

## Results

### Characterization of neural rosette emergence in well plate culture

Forebrain NECs were derived from hESCs using our previously published E6 protocol (*Lippmann et al., 2014*). In six days and without the use of SMAD inhibitors (*Chambers et al., 2009*), the E6 protocol generates near homogenous monolayers of Pax6$^+$/Sox2$^+$/N-cadherin$^+$ NECs that spontaneously produce cultures densely populated with neural rosettes of varying shapes and sizes (*Figure 1A*). To better characterize neural rosette emergence, we conducted a time course analysis using Pax6 and N-cadherin immunostaining (*Figure 1B*). In agreement with previous studies (*Lippmann et al., 2014*), the onset of Pax6 expression, a human neuroectodermal fate determinant (*Zhang et al., 2010b*), began between days 2 (0%) and 3 (44 ± 15%) of the E6 protocol (*Figure 1C–D*). By day 3 and concurrent with increasing Pax6 expression, the presence of polarized N-cadherin$^+$ foci were observed. By day 5, Pax6 expression had reached 81 ± 5% and N-cadherin$^+$ polarization foci had both increased in prevalence and morphed into coherent rings surrounded by contiguous alignment of polarized NECs. Co-localization of F-actin, ZO-1 tight junction proteins, and phospho-H3$^+$ mitotic nuclei within these polarized structures further supports the use of polarized N-cadherin as a surrogate marker of neural rosette emergence (*Figure 1—figure supplement 1A*) *Shi et al., 2012*. Thus, neural rosette emergence occurs over five days after commencing E6 culture in 6-well plates with correlated and progressive increases in Pax6 expression and N-cadherin polarization.

### Characterization of neural rosette emergence time course within micropatterned tissues

Hundreds of neural rosettes form spontaneously and randomly upon NEC derivation in vitro (*Figure 1A*). We hypothesized that the prevalence of neural rosettes could be regulated by controlling tissue size and geometry. This was corroborated by our previous observation that ~0 – 4 neural rosettes routinely form within micropatterned, 300 µm diameter circular NEC tissues (*Knight et al., 2015*). Based on the temporal dynamics of rosette formation observed in well plate culture (*Figure 1B–D*), we first characterized the timeline of neural rosette emergence within micropatterned neural tissues. NECs were harvested on day 4 of the E6 protocol (D4 NECs), which corresponded with increasing Pax6 expression and rosette formation capacity (*Figure 1E*). Next, they were seeded at 75,000 cells/cm$^2$ onto micropatterned, poly(ethylene glycol methyl ether)-grafted substrates presenting arrays of 300 µm diameter circular regions (0.071 mm$^2$) coated with Matrigel (*Figure 1—figure supplement 1B*) (*Knight et al., 2014*; *Sha et al., 2013*). Time course analysis over the next 8 days using Pax6 and N-cadherin immunostaining revealed several distinct stages of rosette emergence that we classified as pre-polarization (~day 7), polarization foci (~day 8), neural rosette (~day 9), and contraction (~day 13) (*Figure 1F*). Interestingly, despite being D4 NECs upon seeding onto micropatterned substrates,~5 days of additional E6 culture was still required for neural rosette emergence similar to well plate culture (*Figure 1C*). The 5 day time point also correlated with the highest occurrence (30%, n = 100 tissues) of singular rosette emergence within micropatterned NEC tissues (*Figure 1G*). Additionally, the NECs appeared to proliferate within micropatterned tissues after seeding as indicated by increased Pax6$^+$ cells density (e.g. day 7 vs 9, *Figure 1F*).

Persistence of the approximate 5 day time course for neural rosette emergence in E6 media indicates that temporal aspects of rosette emergence may rely on a conserved aspect of E6 culture. As a final test of the 5 day paradigm, we investigated whether seeding Pax6$^+$ cells on micropatterned substrates was requisite for adherence to the 5 day rosette emergence time course. Neurally differentiating cultures were subcultured at day 1, 2, 3, 4, and 5 of the E6 protocol, that is Pax6 expression being absent in D1-2 and present in D3-5 cultures. The cells were re-seeded onto 300 µm diameter (0.071 mm$^2$) circular micropattern arrays. Cultures were maintained in E6 media for 5 days prior to fixation and immunostaining (*Figure 1H*). In all conditions, the cells proliferated and produced neuroepithelial tissues with at least a 2-fold increase in cell density, >80% Pax6 expression, and consistent neural rosette formation (*Figure 1I* and *Figure 1—figure supplement 1C–D*). Of note, all tested cell phenotypes were observed to initially adhere as monolayers in well plates or on micropatterned substrates even when the seeding density was increased by several folds (data not shown). These results indicate that a 5 day paradigm for neural rosette emergence in E6 media holds

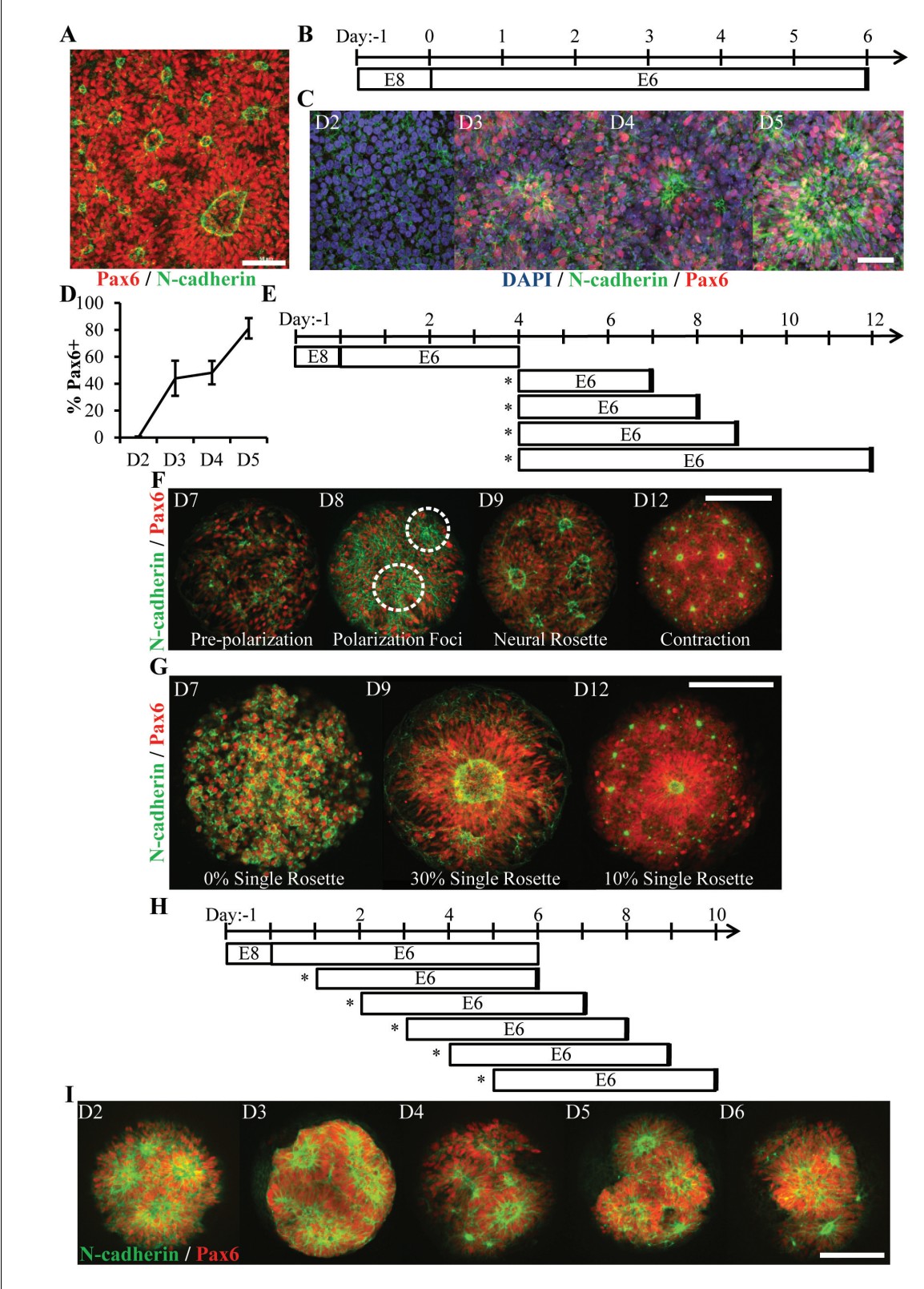

**Figure 1.** Characterization of Neural Rosette Emergence. (**A**) Image of N-cadherin polarized neural rosettes structures within neuroepithelial tissues derived from hPSCs using the E6 method in 6-well TCPS plates. (**B**) Schematic of NEC derivation via the E6 method and (**C**) representative immunostaining images of neural induction (Pax6) and polarization (N-cadherin) time course. Co-staining of neural rosettes with F-actin, ZO-1 tight junction proteins, and phospho-H3[+] mitotic nuclei are provided in *Figure 1—figure supplement 1*. (**D**) Quantification of Pax6 expression during NECs

*Figure 1 continued on next page*

*Figure 1 continued*

derivation (n = 6 fields of view for one well per time point). Error bars represent standard deviation. (E) Schematic of neural rosette emergence within micropatterned tissues derived from D4 NECs; sub-culture/seeding onto micropatterned substrates indicated by (*). (F) Representative images depicting stages of rosette emergence and (G) rate of singularity within neuroepithelial tissues on select days. Dotted lines highlight polarization foci. (H) Schematic of experiment testing the effect of E6 subculture time point on neural rosette emergence within micropatterned neuroepithelial tissues, and (I) corresponding images of immunostained tissues 5 days after culture on micropatterned substrates. Analysis of the tissues' cell density and percentage of Pax6+ cells are provided in *Figure 1—figure supplement 1*. Scale bars are (A, C) 50 µm and (F, G, and I) 200 µm.

DOI: https://doi.org/10.7554/eLife.37549.002

The following source data and figure supplement are available for figure 1:

**Source data 1.** Imaging and quantitative analysis of micropatterned neuroepitelial tissues.
DOI: https://doi.org/10.7554/eLife.37549.004

**Figure supplement 1.** Imaging and Quantitative Analysis of Micropatterned Neuroepithelial Tissues.
DOI: https://doi.org/10.7554/eLife.37549.003

for cultures seeded at all stages of hESC neural induction and appears to be a function of Pax6 expression and local cell density. From here onward, a 5 day micropatterned culture period was used to assess neural rosette emergence.

## Neural rosette emergence is regulated by tissue morphology

The previous experiments demonstrated that switching from a 9.6 cm$^2$ (a well in a 6-well plate) to a 0.071 mm$^2$ (300µm diameter) circular neuroepithelial tissue can substantially reduce the number of emerging neural rosettes. However, formation of neuroepithelial tissues with a biomimetic, singular rosette was not reproducible (30%, *Figure 1G*). Biophysical studies have inextricably linked developmental morphogenesis events such as neural tube formation with tissue biomechanics (*Davidson and Keller, 1999*; *Nishimura et al., 2012*). Specifically, studies on micropatterned cultures have observed that multicellular tissues contract as a unit generating gradients of mechanical stresses with maximal cytoskeletal forces arising at their periphery and in spatial distributions dependent upon tissue morphology (*Nelson et al., 2005*). Therefore, we hypothesized that the reproducibility of singular rosette emergence could be further improved by varying neuroepithelial tissue morphology, that is size and geometry.

This hypothesis was tested by assessing singular rosette induction efficiency within micropatterned neuroepithelial tissues of circular, triangle, square, and oblong morphologies of varying sizes (*Figure 2A*). Culture substrates were designed to normalize the surface area of all tissue morphologies of a given size scale to 200 (0.031mm$^2$), 300 (0.071mm$^2$), or 400µm (0.126mm$^2$) diameter circular micropatterns. In this manner, the cell seeding density would be constant amongst tissues of differing morphologies within the same size scale. Day 4 NECs were seeded onto 0.031, 0.071, and 0.126mm$^2$ micropattern arrays at 33,300, 75,000, and 133,000 cells/cm$^2$, respectively, to correspond with changes in micropattern cell-adhesive area across each size scale. After 5 days in E6 media, the cultures were fixed, immunostained for Pax6 and N-cadherin, and analyzed manually to quantify the number of N-cadherin+ polarization foci and rosettes (*Figure 2B*). Each tissue was categorized as follows: the presence of 0 or one polarization foci were '0 Rosette'; 1 rosette and 0 polarization foci were '1 Rosette'; ≥1 rosette with ≥1 additional polarization foci were '+1 Rosette'. Although all micropatterned tissue morphologies exhibited the capacity to induce '1 Rosette' neuroepithelial tissues, 0.031mm$^2$ circular (56%) and square (57%) morphologies were the most efficient. Yet, manual analysis of each image limited our ability to screen larger sample sizes per tissue morphology.

To increase sample size, an automated workflow consisting of an image analysis algorithm and machine learning classifier was developed to quantitatively characterize and classify N-cadherin+ polarization foci and rosettes within micropatterned tissues (*See Materials and methods*, *Figure 2C*, and *Figure 2—figure supplement 1*). The automated workflow was used to analyze a total of 1633 images of neuroepithelial tissues with circular, triangle, and square morphologies across 0.031, 0.071, and 0.126 mm$^2$ surface area size scales. Compared to manual assessment, image analysis using the automated workflow estimated lower percentages of singular rosette emergence efficiency across all tissue morphologies (*Figure 2D*). This was expected since the classifier's rosette identification error rate is primarily due to false positives, that is erroneously classifying a polarization foci as rosette (Table S2). However, the trends in single rosette emergence efficiency were consistent with

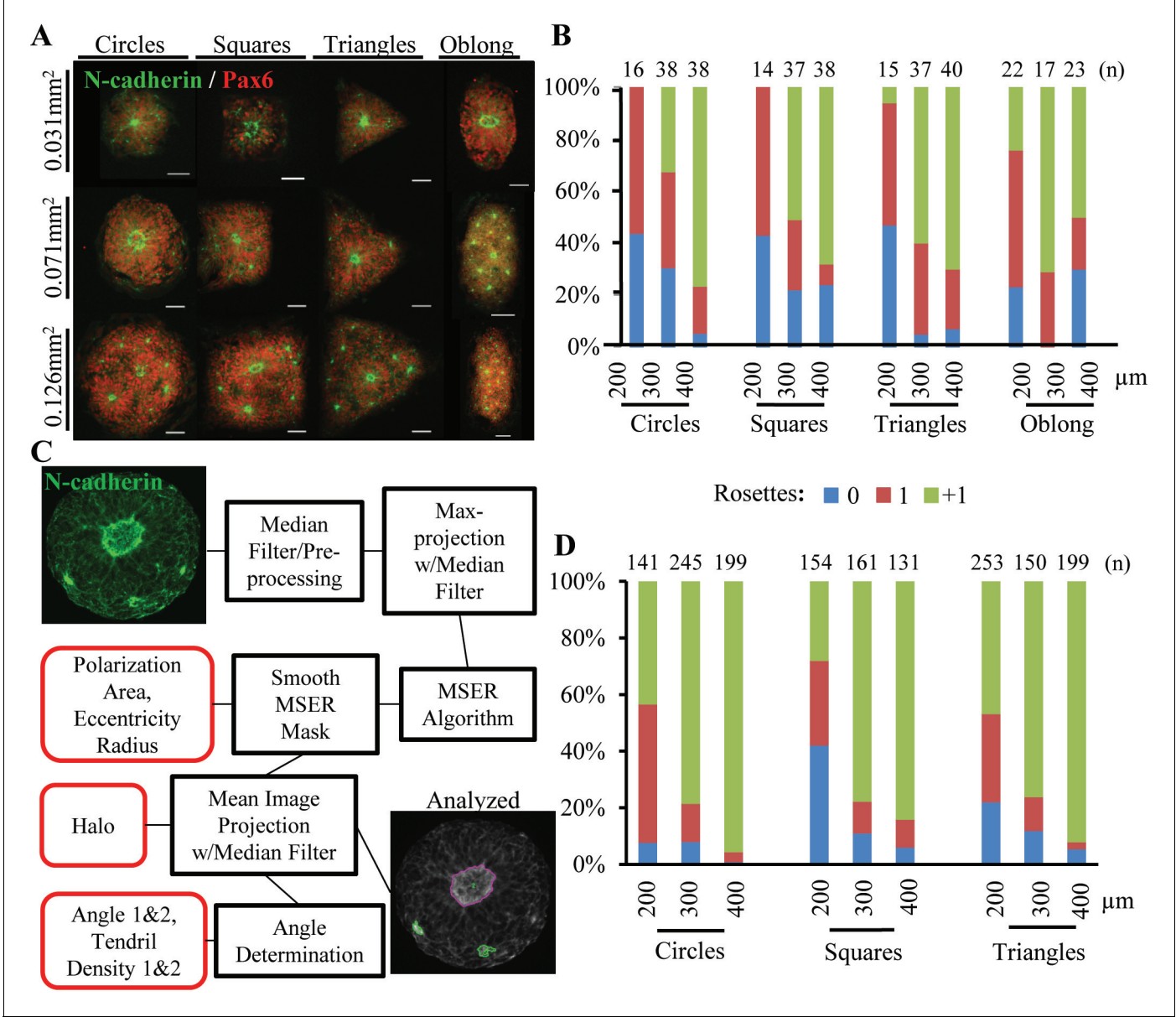

**Figure 2.** Effect of Tissue Morphology on Singular Rosette Emergence. (A) Representative images of micropatterned neuroepithelial tissues of circle, square, triangle, and oblong morphologies of varying size. Scale bars are 50 µm. (B) Manual quantification of polarization foci and neural rosettes within micropatterned neuroepithelial tissues of varying morphology. Number of tissues (technical replicates) analyzed per geometry is indicated above each bar. (C) Schematic of automated image analysis algorithm for quantification of polarization foci and neural rosettes. In 'Analyzed' micrograph, the purple outline indicates a rosette and green outlines indicate polarization foci as determined by the image analysis workflow. Micrograph's providing further visual representation of neuroepithelial tissue descriptors (outlined in red) detected by the image analysis algorithm are provided in *Figure 2— figure supplement 1*. (D) Automated quantification of polarization foci and neural rosettes in micropatterned neuroepithelial tissues of varying morphology. Number of tissues (technical replicates) analyzed per geometry is indicated above each bar.

DOI: https://doi.org/10.7554/eLife.37549.005

The following source data and figure supplement are available for figure 2:

**Source data 1.** Manual and automated quantification of neural rosette emergence.
DOI: https://doi.org/10.7554/eLife.37549.007

**Figure supplement 1.** Schematic of Automated Image Analysis Algorithm Descriptors.
DOI: https://doi.org/10.7554/eLife.37549.006

manual assessment in *Figure 2B*. The probability of biomimetic, singular rosette emergence increased as the neuroepithelial tissue's surface area decreased. Interestingly, the automated analysis of increased sample size suggests that 200 μm diameter (0.031 mm$^2$) circular versus square or triangle morphologies is more efficient at inducing singular rosette emergence, that is 48.9% vs. 29.9% vs. 31.2% respectively (*Figure 2D*).

## Seeding at the hESC state increases singular rosette emergence with forebrain neuroepithelial tissues

In *Figure 1*'s rosette emergence characterization experiments, it was also noted that tissues formed using Pax6$^-$ D1-2 cultures versus Pax6$^+$ D3-5 cultures displayed a significant increase in neuroepithelial cell density (*Figure 1—figure supplement 1C*). Thus, we hypothesized that rosette emergence could be affected by the micropatterned cells' proliferative capacity over 5 days of E6 culture. This was tested using two disparate cell phenotypes encountered during E6 neural induction, that is Pax6$^-$ hESCs (D$^-$1) and Pax6$^+$ D4 NECs (*Figure 3A*). Both cell types were seeded onto 200 μm diameter (0.031 mm$^2$) circular micropattern arrays in biological duplicate, fixed, and immunostained after 5 days (*Figure 3B*). In all tissues, the cells obtained a Pax6$^+$/Otx2$^+$ dorsal forebrain phenotype (*Lippmann et al., 2014*) with near uniformity. Manual analysis of over 100 neuroepithelial tissues in each experiment revealed a singular neural rosette emergence efficiency of 80.0 ± 0.05% vs. 38.6 ± 4.7% for neuroepithelial tissues formed by seeding D$^-$1 hESCs vs. D4 NECs, respectively (*Figure 3C*). The average cell density of neuroepithelial tissues formed using D$^-$1 hESCs vs. D4 NECs was ~1.25 fold higher after 5 days of E6 culture or ~30 more cells per tissue, that is 149 ± 23 vs. 119 ± 18 cells respectively (*Figure 3D*). Thus, E6 culture of hESCs on 200 μm diameter (0.031 mm$^2$) circular micropatterns induces reproducible, singular rosette emergence within forebrain neuroepithelial tissues in part due to their increased proliferative capacity. Additionally, although all tissues were multilayered, the differences in cell density correlated with differences in neuroepithelial tissue morphology. Singular rosettes tissues derived from D$^-$1 hESCs exhibited apico-basal polarity from a wider, central N-cadherin ring, while D4 NEC-derived singular rosette tissues had a tighter N-cadherin ring and appeared to have contracted slightly from their initial boundary (*Figure 3B,E and F*).

## Controlled induction of singular neural rosettes within spinal tissues

The NEC polarized morphology is consistent throughout all regions of the primordial neural tube. However, their gene expression profile induced by morphogenetic patterning varies along both the neural tubes' rostrocaudal (R/C) (*Philippidou and Dasen, 2013*) and dorsoventral (D/V) (*Briscoe et al., 2000*; *Timmer et al., 2002*) axes. NECs derived using the E6 protocol pattern to a default Pax6$^+$/Otx2$^+$ dorsal forebrain phenotype both in well plate and micropatterned tissue culture (*Figure 3B*). We have also published a protocol for deriving NECs of discrete hindbrain through spinal cord regional phenotypes via deterministic patterning of their *HOX* expression profiles (*Lippmann et al., 2015*) (*Figure 4A*). The protocol begins with high density seeding of hPSCs (D$^-$1) followed by induction of a stable neuromesodermal progenitor (NMP) phenotype using sequential activation of FGF8b and Wnt signaling. Upon addition of the Wnt agonist (CHIR), *HOX* expression activates in a colinear and combinatorial manner over 7 days. At any time point within those 7 days, the NMPs can be differentiated into NECs by transitioning from a FGF8b/CHIR to a retinoic acid (RA) supplemented E6 media. Based on the time point at which this transition is made, the resulting NECs will express a distinct *HOX* profile indicative of phenotypical patterning to a discrete hindbrain through spinal cord R/C region. To enable reproducible derivation of biomimetic neural tissues from forebrain through spinal cord regions, we investigated how to integrate the *HOX* patterning protocol with the micropatterning methodology for inducing singular neural rosette emergence.

The mitogenic properties of FGF8 and Wnt prevent direct transfer of the *HOX* patterning protocol to D$^-$1 hESC-seeded micropatterned substrates. After only a few days, the cells would proliferate to such a great extent that they would form spheroids that detached from the culture surface. Given our prior positive results with seeding Pax6$^-$ cells (*Figure 1H–I* and *Figure 3*), we explored whether seeding Pax6$^-$/Sox2$^+$/T$^+$ NMPs instead would still permit robust neural induction and singular rosette emergence. In this manner, the NMP culture could be caudalized to the desired R/C position using standard 6-well plate culture, and then seeded onto micropatterned substrates concurrent with transitioning to RA supplemented media to generate NECs (*Figure 4B*). For these experiments, we

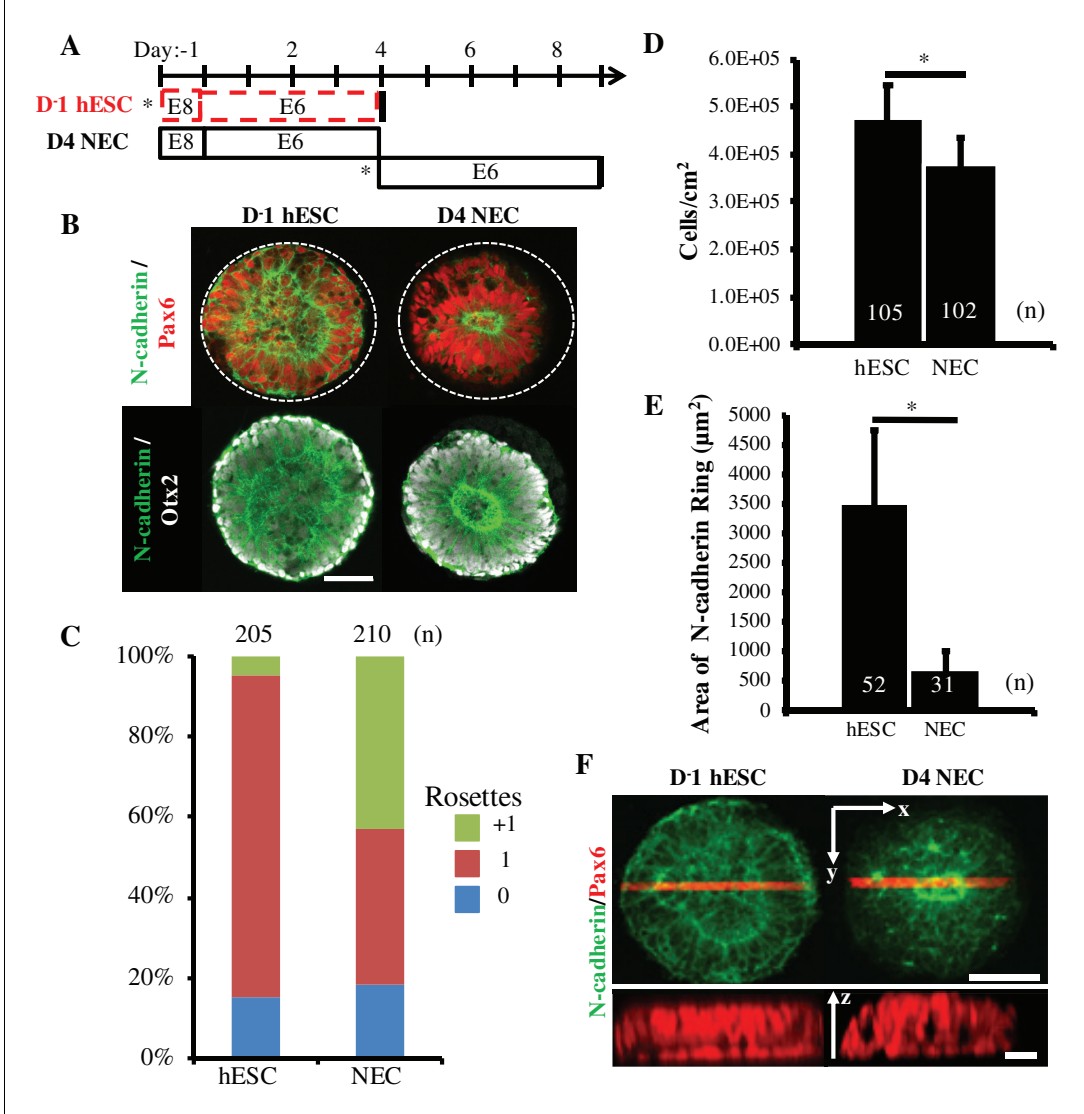

**Figure 3.** Comparison of Rosette Emergence in D⁻1 hESC vs D4 NEC micropatterned tissues. (A) Schematic of micropatterned tissue derivation for D⁻1 hESCs and D4 NECs; sub-culture/seeding onto micropatterned substrates indicated by (*). (B) Representative immunostained images of forebrain, micropatterned neuroepithelial tissues. (C) Manual quantification of polarization foci/neural rosettes per tissue with the number of tissues (technical replicates) analyzed per condition indicated above each bar. (D) Average cell density and (E) area of polarized N-cadherin ring within micropatterned neuroepithelial tissues derived from D⁻1 hESCs and D4 NECs. Number of tissues analyzed per condition indicated on each bar, and error bars represent standard deviation. (*) indicates a significance of p<0.05 calculated using a One-way ANOVA. (F) Representative confocal Z-stack images of micropatterned neuroepithelial tissues (top) with profile view of red highlighted region (bottom). Scale bars are (B, F-top) 100 µm and (F-bottom) 10 µm.

DOI: https://doi.org/10.7554/eLife.37549.008

The following source data is available for figure 3:

**Source data 1.** Quantification of rosette emergence, tissue cell density, and polarized N-cadherin ring area.

DOI: https://doi.org/10.7554/eLife.37549.009

derived spinal NECs using a 72 hr caudalization (i.e. CHIR exposure) period, which is known to pattern a lower brachial cervical/thoracic regional phenotype (*Lippmann et al., 2015*). Regional patterning was confirmed by qRT-PCR (*Figure 4—figure supplement 1A*). Additionally, the ability to induce dorsoventral patterning with a transient spike in Wnt (CHIR) followed by sustained Sonic Hedgehog (SHH) signaling during neural induction was also tested. Both the transient Wnt

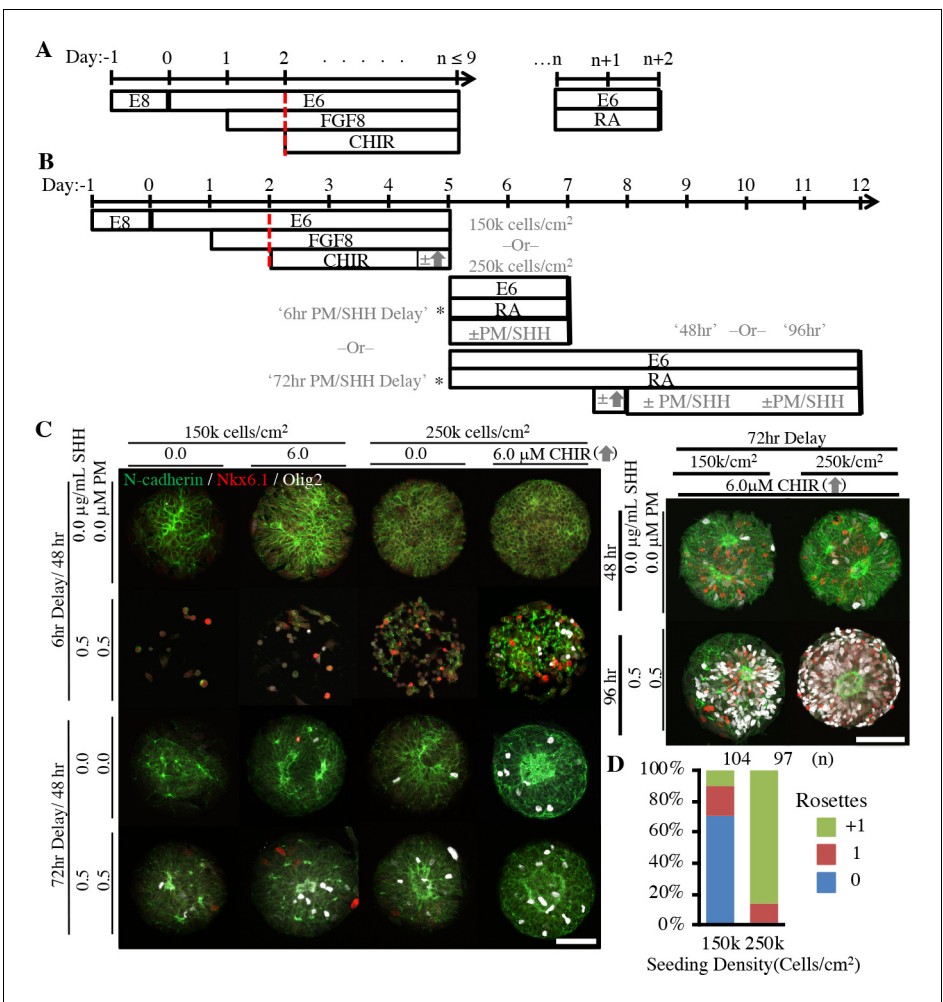

**Figure 4.** Derivation of micropatterned spinal neuroepithelial tissues. (**A**) Schematic for deterministic *HOX* patterning of hPSC-derived neuroepithelium in well plate culture. Dotted red bars indicated well-plate subculture. (**B**) Schematic for translation of well plate protocol to micropatterned arrays with sub-culture (*) of cervical *HOX* patterned NMPs onto micropatterned substrates and subsequent differentiation to ventral spinal tissues. Tested variables indicated in gray. (**C**) Representative images of tissues derived when varying seeding density, timing of CHIR boost, and timing and duration of ventral patterning using Sonic Hedgehog (SHH/PM) signaling. Scale bars are 100 μm. Analysis of spinal tissue *HOX* expression plus quantification of Nkx6.1/Olig2 expression and cell density in each experimental condition is provided in *Figure 4—figure supplement 1*. (**D**) Quantification of polarization foci/rosettes per tissue with the number of tissues (technical replicates) analyzed per condition above each bar.

DOI: https://doi.org/10.7554/eLife.37549.010

The following source data and figure supplement are available for figure 4:

**Source data 1.** Quantification of micropatterned spinal neuroepithelial tissue regional patterning and cell density.
DOI: https://doi.org/10.7554/eLife.37549.012

**Figure supplement 1.** Characterization of regional specification of micropatterned spinal neuroepithelial tissues.
DOI: https://doi.org/10.7554/eLife.37549.011

---

(*Nordström et al., 2006*) and sustained SHH (*Dessaud et al., 2007*) signaling are known regulators of ventral spinal patterning in vivo.

Using a combinatorial design, several experimental variables were tested with spinal NMPs on 200 μm diameter (0.031 mm$^2$) circular micropatterns (*Figure 4B–C*). First, NMPs were seeded at either 1.5 or 2.5 × 10$^5$ cells/cm$^2$ onto micropatterned substrates on Day 5. Second, a transient 6 hr, 6 μM Wnt (CHIR) boost was omitted or applied immediately prior to seeding NMPs onto

micropatterned substrates on Day 5 or after 72 hr of micropatterned culture (i.e. Day 8). Third, ventral patterning with SHH and Purmorphamine (PM), a SHH agonist, was omitted or applied 6 or 72 hr after seeding NMPs onto micropatterned substrates (i.e. Day 5) and for a duration of either 48 or 96 hr. Rosette formation and ventralization were assessed by N-cadherin and Nkx6.1/Olig2, co-markers of spinal motor neuron progenitors, immunostaining respectively (*Lippmann et al., 2014*) (*Figure 4C*). Under the tested conditions, positive immunostaining for floor plate markers FoxA2 and SHH was not observed (data not shown). Manual tissue analysis in each condition revealed that the optimal protocol regimen consisted of seeding spinal NMPs at $2.5 \times 10^5$ cells/cm$^2$ onto micropatterned substrates, culturing in RA supplemented E6 media for 72 hr, and applying a 6 hr, 6.0 µM Wnt boost immediately prior to 96 hr of 0.5 µg/mL SHH and 0.5 µM PM exposure. Waiting 72 hr after NMP seeding before applying the Wnt boost followed by SHH/PM exposure was the only tested regimen that consistently yielded both good cell viability and tissue formation as well as effective ventral patterning, that is 86 ± 11% (n = 8) Nkx6.1$^+$/Olig2$^+$ (*Figure 4C* and *Figure 4—figure supplement 1B–C*). Yet, only ~12% (n = 97) of neuroepithelial tissues on 200 µm diameter (0.031 mm$^2$) circular micropatterns displayed singular rosette emergence (*Figure 4D*).

## NEC biomechanical properties vary by regional phenotype and affect rosette emergence behavior

The inefficient induction of singular rosette emergence in spinal neuroepithelial tissues under micropatterning conditions that were efficient for forebrain NECs prompted a re-examination of the optimal circular micropattern size for each regional phenotype. Moreover, we also investigated how inhibition of Rho-associated protein kinase (ROCK)-mediated actomyosin contraction affects which micropattern size is optimal for singular rosette emergence, since ROCK plays a key role in the planar cell polarity pathway that induces neural tube formation in vivo (*Nishimura et al., 2012*). D⁻1 hESCs and 72 hr caudalized NMPs were seeded at 100,000 and 250,000 cells/cm$^2$, respectively, onto 150 (0.018mm$^2$), 180 (0.025mm$^2$), 200 (0.031mm$^2$), 250 (0.049mm$^2$), and 400 µm (0.126 mm$^2$) diameter circular micropattern arrays to generate forebrain and spinal neuroepithelial tissues (*Figure 5A–B and D–E*). After 5 days of culture in the absence or presence of 10 µM ROCK inhibitor, manual analysis of rosette emergence revealed stark differences in the emergent behavior of forebrain vs. spinal neuroepithelial tissues. Similar to our previous results, the efficiency of singular neural rosette emergence for forebrain tissues peaked at 85% on 250 µm (0.049 mm$^2$) diameter circular micropatterns (*Figure 5C*). However, in the presence vs. absence of ROCK inhibitor, the efficiency curve shifted leftward inducing singular rosette emergence in 75% vs. 0% of 150 µm (0.018 mm$^2$), 89% vs. 50% of 180 µm (0.025 mm$^2$), 87% vs. 58% of 200 µm (0.031 mm$^2$), and 11% vs. 85% of 250 µm (0.049 mm$^2$) diameter forebrain neuroepithelial tissues, respectively. In comparison, singular neural rosette emergence within spinal neuroepithelial tissues peaked at 73.5% on 150 µm (0.018 mm$^2$) diameter micropatterns, and the presence of ROCK inhibitor completely abolished their ability to form neural rosettes (*Figure 5F*). Of note, NEC density within forebrain vs. spinal tissues of a given micropattern size and in the presence vs. absence of ROCK inhibitor did not vary significantly (*Figure 5—figure supplement 1*).

Considering our previous experiment (*Figure 3*), the optimal morphology for single neural rosette emergence in forebrain and spinal neuroepithelial tissues is 200–250 µm (0.031–0.049 mm$^2$) and 150 µm (0.018 mm$^2$) diameter circular micropatterns respectively. The physical size of emergent neural rosettes is not conserved across tissues of increasing area (*Figure 5B and E*). Hence, the micropatterned tissue morphology does not instruct rosette morphogenesis, but restricts tissue area to statistically favor emergence of a single rosette in accordance with innate NEC properties. Since ROCK inhibition down-regulates actin polymerization and actomyosin contraction (*Amano et al., 2010*), the results in *Figure 5* imply that decreasing the contractility of forebrain NECs reduces the length scale over which the cells can effectively polarize to evoke a rosette structure, that is 250 µm vs. 150/180/200 µm in 0 vs. 10 µM Rock inhibitor. This further suggests that forebrain vs. spinal NECs can generate more contractile force since singular rosette emergence peaks in larger forebrain, that is 250 µm (0.049 mm$^2$) diameter, versus smaller spinal, that is 150 µm (0.018 mm$^2$) diameter, circular tissues. The ability of 10 µM Rock inhibitor to only partially diminish vs. fully eliminate the rosette formation capacity of forebrain vs. spinal NECs also supports this ranking of contractility. These results provide the first evidence that NEC biomechanical properties vary based on their regional patterning. They also demonstrate the necessity of customizing the micropatterned

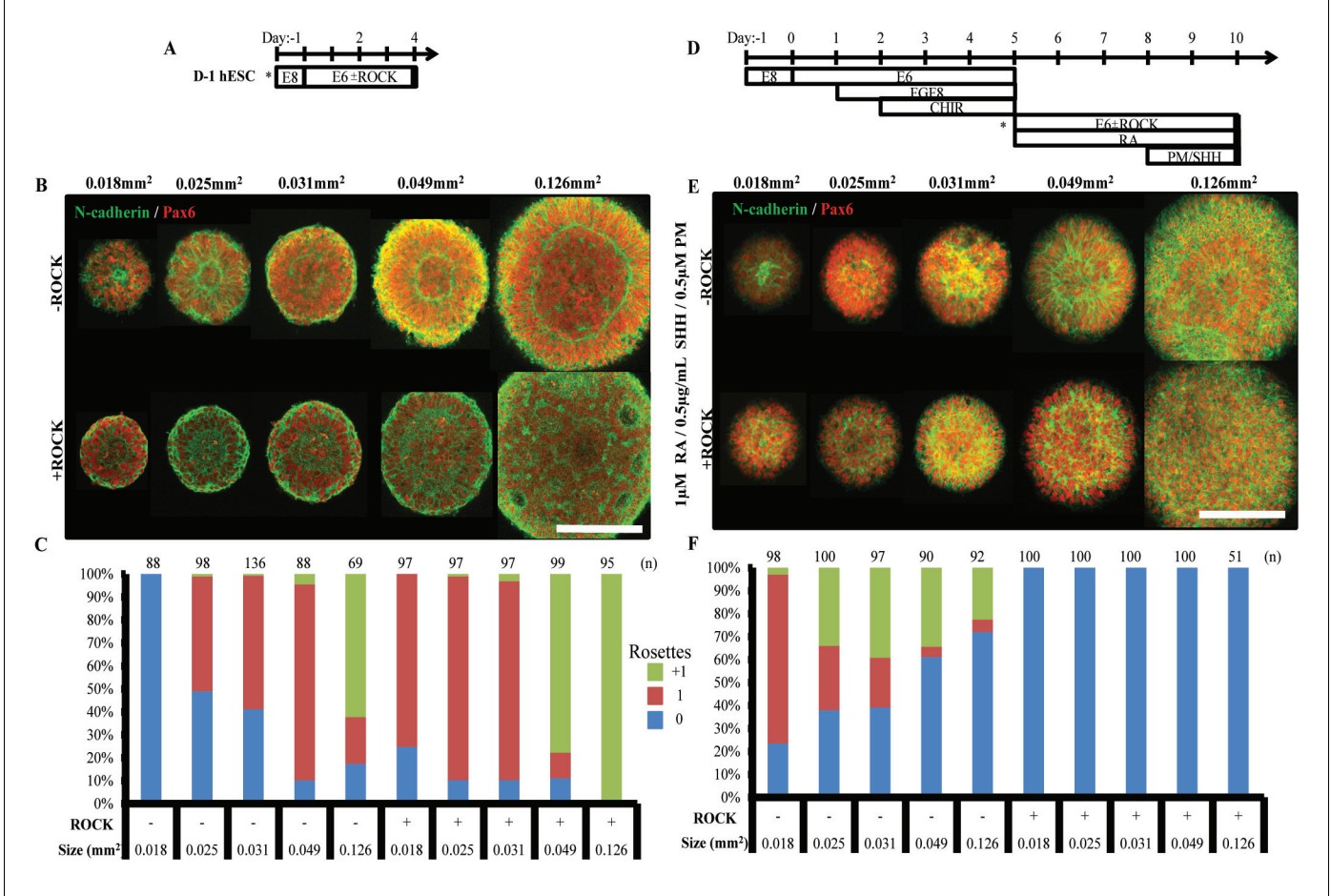

**Figure 5.** Singular neural rosette emergence within forebrain and spinal neuroepithelial tissues. (A) Schematic for derivation of micropatterned forebrain neuroepithelial tissues; subculture/seeding onto micropatterned substrates indicated by (*). (B) Representative images of neural rosette emergence in micropatterned, hESC-derived forebrain neuroepithelial tissues of various areas with and without ROCK inhibitor. (C) Quantification of polarization foci/rosettes per forebrain tissue with the number of tissues analyzed per condition above each bar. (D) Schematic for derivation of micropatterned spinal neuroepithelial tissues; sub-culture/seeding onto micropatterned substrates indicated by (*). (E) Representative images of neural rosette emergence in micropatterned, hESC-derived spinal cord neuroepithelial tissues of various areas with and without ROCK inhibitor. (F) Quantification of polarization foci/neural rosettes per spinal tissue with the number of tissue (technical replicates) analyzed per condition above each bar. *Figure 5—figure supplement 1* provides quantification of average cell density within hESC-derived forebrain and spinal neuroepithelial tissues for each experimental condition. *Figure 5—figure supplement 2* shows a representative image and quantification of polarization foci/rosettes per forebrain tissue derived from D90A *SOD1* mutant ALS-iPSCs. Scale bars are (B, F) 200 μm.
DOI: https://doi.org/10.7554/eLife.37549.013

The following source data and figure supplements are available for figure 5:

**Source data 1.** Quantification of cell density and rosette emergence in forebrain and spinal and iPSC-derived tissues.
DOI: https://doi.org/10.7554/eLife.37549.016

**Figure supplement 1.** Cell density analysis of forebrain and spinal cord neuroepithelial tissues.
DOI: https://doi.org/10.7554/eLife.37549.014

**Figure supplement 2.** Singular rosette emergence within D90A *SOD1* mutant ALS-iPSCs.
DOI: https://doi.org/10.7554/eLife.37549.015

neuroepithelial tissue morphology to efficiently induce singular rosette emergence based on the cell's regional phenotype.

# Singular neural rosette cytoarchitecture is maintained during radial tissue outgrowth

The importance of maintaining a singular neuroepithelium in early CNS development is evident given severe congenital disorders associated with breakdown (*Qian et al., 2016*; *Watanabe et al., 2017*) or duplication (*Testoni et al., 2010*; *Spencer, 2000*) of the neuroepithelial tube. Accordingly, we investigated whether the singular rosette structure induced within micropatterned neuroepithelial tissues would persist upon subsequent tissue growth and morphogenesis. Using our previously developed 'clickable' polyethylene glycol (PEG) brush chemistry (*Knight et al., 2015*; *Sha et al., 2013*), we synthesized array substrates that could be actuated to release the micropatterned tissues from their confinement and permit radial tissue outgrowth. In our prior publication, tissue outgrowth was enabled by in situ conjugation of –RGD peptide bioconjugates to the microarray's PEG brushes. However, this caused aberrant migration of single cells away from the neuroepithelial tissues instead of a more coordinated, biomimetic radial tissue expansion (*Knight et al., 2015*). Therefore, we explored whether a clickable bioconjugate containing a heparin-binding peptide (-CGTYRSRKY) derived from Fibroblast growth factor-2 would mediate a more gradual tissue expansion. This was posited based on the fact that the heparin-binding peptide (HBP) is unable to directly promote cell migration via integrin binding but could immobilize heparin sulfate proteoglycan-bound ECM proteins (e.g. Laminin) secreted at the neuroepithelial tissues' basal aspect upon extended culture (*Hudalla and Murphy, 2010*) (*Figure 6—figure supplement 1*).

To evaluate feasibility of our proposed substrate design, micropatterned arrays with reactive, 200 µm (0.031 mm$^2$) diameter, circular PEG brushes were fabricated and functionalized via an overnight incubation in cell culture media containing 0 to 20 µM FITC-HBP conjugated to a Malemide-PEG4-DBCO linker (*Figure 6—figure supplements 2* and *3A–B*). Using confocal microscopy, a direct correlation between fluorescence from immobilized FITC-HBP and the incubation media concentration was observed. Culture media containing 20 µM FITC-HBP-DBCO bioconjugate provided the maximal surface density. Then, we tested whether HBP-presenting PEG brushes could sequester ECM proteins. Substrates modified with cell culture media containing 0, 5, 10, and 20 µM HBP-DBCO bioconjugate were subsequently incubated overnight in media containing 10 µM NHS-rhodamine tagged Laminin, which was isolated from a mouse sarcoma and likely co-bound with heparin sulfate proteoglycans. Imaging revealed a consistent increase in rhodamine fluorescence directly correlated with increasing surface densities of HBP. The quantified fluorescent signal saturated on substrates modified with culture media containing 10 µM HBP-DBCO bioconjugate (*Figure 6—figure supplement 3C–D*). Thus, PEG-brushes modified in situ with 10 µM HBP-DBCO bioconjugate are capable of immobilizing Laminin proteins, and potentially a diverse cadre of basement membrane ECM proteins that bind heparin sulfate proteoglycans (*Sarrazin et al., 2011*).

Next, neuroepithelial tissue outgrowth was tested on both inert and reactive PEG-grafted substrates micropatterned to display arrays of 200 µm (0.031 mm$^2$) diameter circular culture regions with 400 µm center-to-center spacing. The substrates were coated with Matrigel to deposit ECM proteins on the 200 µm diameter culture regions. Pax6$^-$ D$^-$1 hESCs were seeded onto the substrates at 100,000 cells/cm$^2$ and maintained in E6 culture for 3 days, at which point the media was supplemented with 10.0 µM HBP-DBCO bioconjugate for 24 hr on both inert and reactive microarray substrates. After three additional days of culture, the tissues were fixed and stained for N-cadherin and Laminin (*Figure 6A*). Inert control PEG brushes predictably maintained the restricted singular rosette tissue morphology for the culture duration (*Figure 6B*). In contrast, on reactive substrates modified in situ with HBP-DBCO bioconjugates, the tissues expanded as a coherent, multilayered structures maintaining a singular neuroepithelium and eventually intersecting with neighboring tissues. Also, they displayed basal deposition of laminin resembling the developing neural tube's basal lamina (*Figure 6C–D* and *Figure 6—figure supplement 4*). To prevent inhibition of tissue outgrowth, the experiment was repeated on reactive micropatterned arrays with 200 µm (0.031 mm$^2$) diameter circular regions with larger 800 µm center-to-center spacing. The in situ modification occurred on Day 4 of E6 culture, and microarrays were fixed and stained after an additional 7 days of culture (*Figure 6E*). These tissues also maintained a singular Pax6$^+$ neuroepithelium throughout the culture period that morphed into a 3-D hemispherical structure with a central cavity and Tuj1$^+$ neuronal cells residing at the tissue's periphery (*Figure 6F–G* and *Figure 6—video 1*). These results indicate that controlled induction of singular rosette emergence upon organotypic neural tissue formation will

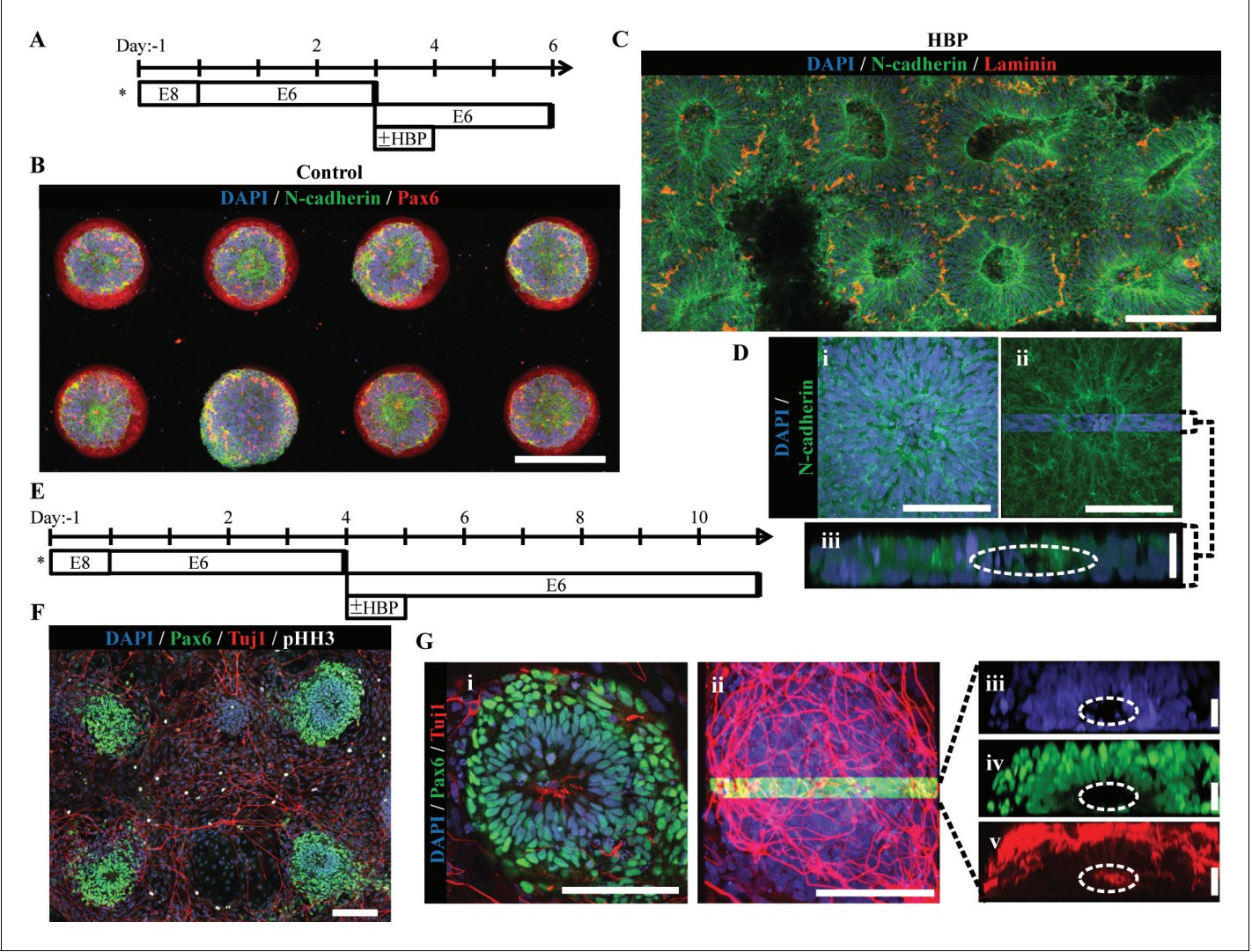

**Figure 6.** Micropatterned neuroepithelial tissue outgrowth on clickable, HBP-presenting substrates. (A) Schematic for derivation and HBP-mediated expansion of forebrain neuroepithelial tissues on chemically-modified micropattern arrays of 200 μm diameter circles with 400 μm center-to-center spacing. *Figure 6—figure supplements 2* and *3* show spectroscopic and protein assay quantification of HBP-DBCO bioconjugates and functional demonstration in their surface immobilized form, respectively. (B) Image of singular neural rosette emergence within D6 neuroepithelial tissues on inert PEG-MA substrates not functionalized by HBP-DBCO bioconjugates. (C) Image of singular neural rosette expansion within D6 neuroepithelial tissues on reactive PEG-MA substrates functionalized with HBP-DBCO bioconjugates. The tissues show radial expansion and deposition of a Laminin$^+$ basement membrane analogous to extended neuroepithelial culture in well plates as shown in *Figure 6—figure supplement 1*. Bright-field, time course images of radial tissue expansion are shown in *Figure 6—figure supplement 4*. (D) (i) Magnified image of expanded neuroepithelial tissue with (ii) highlighted 3D cross-section and (iii) profile view; white dotted line indicates hollow polarization cavity. (E) Schematic for derivation and HBP-mediated expansion of forebrain neural tissues on chemically-modified micropattern arrays of 200 μm diameter circles with 800 μm center-to-center spacing. (F) Image of singular neural rosette tissue outgrowth and neuronal differentiation on HBP-functionalized substrates at day 11. (G) (i) Magnified image of the neuroepithelial center of expanded neural tissues with (ii) the highlighted 3D cross-section and profile view of (iii) nuclei, (iv) Pax6$^+$ core, and (v) primarily peripheral Tuj1$^+$ neuronal processes; white dotted lines indicate hollow polarization cavity. 3D animated deconstruction of an immunostained tissue that further facilitates cavity visualization is provided in *Figure 6—video 1*. Scale bars are (B, C, F) 200 μm, (D(i,ii), G(i,ii)) 100 μm, and (D(iii), G(iii, iv,v)) 20 μm.

DOI: https://doi.org/10.7554/eLife.37549.017

The following video, source data, and figure supplements are available for figure 6:

**Source data 1.** Quantification of micropatterned substrate engineering.
DOI: https://doi.org/10.7554/eLife.37549.023

**Figure supplement 1.** ECM deposition by neuroepithelial tissues upon extended culture.
DOI: https://doi.org/10.7554/eLife.37549.018

*Figure 6 continued on next page*

*Figure 6 continued*

**Figure supplement 2.** Characterization of FITC-HBP-DBCO Bioconjugate Synthesis.
DOI: https://doi.org/10.7554/eLife.37549.019
**Figure supplement 3.** Analysis of HBP-DBCO bioconjugate surface density and ECM adsorption.
DOI: https://doi.org/10.7554/eLife.37549.020
**Figure supplement 4.** Time Course Imaging of Patterned NEC Tissue Outgrowth.
DOI: https://doi.org/10.7554/eLife.37549.021
**Figure 6—video 1.** 3D Deconstruction of Patterned Neural Tissue Immunostaining.
DOI: https://doi.org/10.7554/eLife.37549.022

persist throughout subsequent stages of morphogenesis, thereby enabling formation of a biomimetic nascent CNS tissue cytoarchitecture.

## Discussion

Neural organoid technology has enabled unprecedented in vitro recapitulation of biomimetic, human CNS tissue microenvironments. This is currently being used to develop novel personalized neurological disease modeling platforms (*Lancaster et al., 2013*; *Bershteyn et al., 2017*; *Wells et al., 2016*). However, while impressive in their extent of ex vivo morphogenesis, the spontaneous and cell-intrinsic emergent properties that enable organoid formation also currently limits their wide-scale implementation due to the lack of reproducibility in their macroscale cytoarchitecture and cellular composition. In essence, we now have a glimpse of the neural organoid's potential, but do not understand the processes that govern their morphogenesis well enough to precisely engineer their formation (*Huch et al., 2017*). Here, we have melded organoid technology with a biomaterial culture platform to elucidate biophysical parameters that control the genesis of human neural organoid morphogenesis, that is polarized neural epithelium formation.

Over the past decade, scientists have generated human neural organoids of the cerebral cortex (*Eiraku et al., 2008*), retina (*Nakano et al., 2012*), forebrain (*Lancaster et al., 2013*; *Renner et al., 2017*), midbrain (*Jo et al., 2016*), and cerebellum (*Muguruma et al., 2015*) with both neuronal and glial cellular constituents (*Paşca et al., 2015*). However, a consistent limitation in creating reproducible and biomimetic neural organoid anatomy at the macroscale is the random formation of multiple polarized neuroepithelial regions, a.k.a. rosettes, which can each act as independent morphogenesis centers. Using micropatterned culture substrates, we have demonstrated that enforcing a 200 – 250 μm forebrain and 150 μm spinal, circular, neuroepithelial tissue morphology optimally induces formation of a singular rosette cytoarchitecture with 80 – 85% and 73.5% efficiency respectively (*Figures 3* and *5*). Our results indicate that these biophysical parameters for reproducible induction of a biomimetic, neural tube-like cytoarchitecture are dependent on local cell density, acquisition of Pax6 expression, and cell autonomous biomechanical properties such as contractility. Interestingly, while these results were consistent for tested hESC lines, micropatterned E6 differentiation of D90A *SOD1* mutant ALS-iPSCs also showed optimal singular, forebrain rosette emergence within 250 μm diameter circular tissues but at a lower (26%) efficiency (*Figure 5—figure supplement 2*). This corresponded to many instances where tissues displayed a non-uniform Pax6$^+$ NEC phenotype, indicating that line-to-line variations in neural induction efficiency must be overcome for effective micropatterning of singular neural rosette emergence.

Furthermore, we have shown that the singular neural rosette cytoarchitecture can be maintained upon subsequent tissue morphogenesis. This indicates that initial constraint of tissue morphology in accordance with these identified length scales could be a generalizable approach to engineering human neural organoids with standardized nascent CNS tissue cytoarchitecture. However, we acknowledge that our micropatterned tissue do morph from 2-D monolayers into 3-D hemispherical tissues (*Figure 6*), and therefore, our identified optimal tissue morphology parameters may not be precisely conserved for organoids initially formed as 3-D cell aggregates.

While organoids can be used to generate novel humanoid disease models, their ability to recapitulate developmental morphogenesis ex vivo and within a human genomic background also provides unique opportunities to investigate mechanisms that orchestrate early human development (*Warmflash et al., 2014*; *Etoc et al., 2016*). In efforts to develop a unified approach for inducing

singular neural rosette emergence within neuroepithelium patterned to any CNS rostrocaudal region, we discovered that the biomechanical properties of Pax6$^+$/N-cadherin$^+$ NECs of the dorsal forebrain differed from those of the ventral spinal cord. Forebrain NECs appear to be able to form a singular neural rosette within tissue of larger circular dimensions than spinal NECs because of their increased contractility (*Figure 5*). This would seem to correlate with the significantly larger dimensions of forebrain ventricles versus the spinal cord's central canal, and could represent a novel axis between the biochemical morphogens and biomechanical properties that govern CNS morphogenesis. While our conclusions are based on observations from correlated perturbations of ROCK activity and not direct contractility measurements, our experimental paradigm does provide a unique platform for further definitive analysis of the interplay between biochemical and biomechanical factors that govern this facet of human CNS morphogenesis. To our knowledge, there is no other experimentally tractable means by which such phenomena can be investigated within a human genomic background.

The ability to reproducibly engineer the nascent neuroepithelial cytoarchitecture of neural organoids in a high-throughput arrayed format could have significant implications for future advanced biomanufacture of the neural organoid platform. The 3-D nature of organoids grown in suspension culture provides constituent cells with a biomimetic tissue context. However, suspension organoids grow to millimeters in diameter limiting real-time analysis of internal morphogenesis processes using traditional imaging modalities. While our microarrayed neural tissues initiate as a 2-D monolayer, they quickly become multilayered and definitively 3-D, that is >4 cell layers thick, within 4 days of culture and continue to increase in thickness upon extended culture (*Figures 3* and *6*). Thus, the biophysical (*Khetan et al., 2013*) and electrophysiological (*Frega et al., 2014*) benefit of 3-D organoids may still apply within our engineered neural tube slides. Also, our arrayed approach enables constant monitoring throughout the tissues' morphogenesis process using confocal microscopy. Furthermore, the stratified microscale cytoarchitecture observed in regions of cortical, retinal, cerebellar, and cerebral organoids may also be achievable upon further modification of our culture substrates to provide continual but expanding confinement as the arrayed tissues grow radially (*Knight et al., 2015*; *McNulty et al., 2014*). If achievable, then the substrate parameters elucidated in this study could serve as a basis for generating an engineered well-plate platform to which scientists could simply add their patient-specific hPSCs or Pax6$^-$ hPSC derivatives (*Figures 3* and *4*), follow an in situ modification protocol (*Figure 6*), and allow the engineered culture substrate to instruct reproducible growth of high density arrays of neural organoid tissues. This would facilitate broad dissemination of the neural organoid platform as well as real-time microscopic investigation of the organoid's morphogenesis and terminal structure.

## Materials and methods

### Micropatterned array substrate fabrication

Micropatterned array cell culture substrates were fabricated using a combination of previously published methods (*Knight et al., 2015*; *Sha et al., 2013*). Polydimethylsiloxane (PDMS) stamps with arrays of post and micro-well features were generated as relief molds of silicon wafers. The wafers were designed in AutoCAD and purchased from FlowJEM (www.flowjem.com). PDMS stamps were coated with ω-mercaptoundecyl bromoisobutyrate (2 mM in 100% ethanol), dried under inert gas, and then brought in conformal contact with glass coverslips coated with 180 nm Au atop 30 nm Ti. Micropatterned slides were then incubated in 100% ethanol for 10 min, prior to being dried under nitrogen and transferred to a Schlenk flask under vacuum. A solution of either poly(ethylene glycol) methyl ether methacrylate (PEG-MEMA) or poly(ethylene glycol) methacrylate (PEG-MA) macromonomer (Sigma Aldrich) with water, methanol (Thermo Fisher), copper(II) bromide (Sigma Aldrich), and 2',2-bipyridine (Sigma Aldrich) was degassed and transferred to the reaction flask. Surface-initiated atom-transfer radical-polymerization (SI-ATRP) of PEG polymers was initiated by injection of L-ascorbic acid (Sigma Aldrich) in deionized water into the reaction flask. ATRP was allowed to continue for 16 hr at room temperature to generate micropatterned PEG brushes. Polymerization was terminated via addition of air and followed by rinsing with ethanol and water before drying under inert gas. In a sterile hood, the substrates were rinsed five times with sterile PBS (Thermo Fisher) and transferred to individual wells of a 12-well tissue-culture polystyrene (TCPS) plate where they were

rendered cell-adhesive through adsorption of 0.083 mg/mL Matrigel (WiCell) in DMEM/F-12 (Thermo Fisher) via overnight incubation at 37°C.

## Generation of micropatterned forebrain neuroepithelial tissues

WA09 (H9) and HUES3 Hb9::GFP hESCs were obtained from WiCell and the Harvard Stem Cell Institute, respectively. D90A *SOD1* mutant ALS-iPSCs were obtained from the NINDS Cell Repository. All lines were authenticated as karotypically normal by the provider, used for no more than 15 passages from receipt, and were tested for mycoplasma with negative results (Charles River and WiCell). All pluripotent lines were maintained in Essential eight medium (E8) on Matrigel-coated TCPS plates and routinely passaged with Versene (Thermo Fisher). NEC derivation from hPSCs was performed in accordance with the E6 protocol (*Lippmann et al., 2014*). To generate NEC-derived micropatterned tissues at various stages of neural derivation, cells were first rinsed with PBS, dissociated with Accutase (Thermo Fisher) for 5 min at 37°C, and collected via centrifugation at 1000 rpm for 5 min. Singularized NECs were re-suspended in E6 media with 10 µM ROCK inhibitor (Y27632; R and D Systems) and seeded onto micropatterned substrates at 75,000 cells/cm$^2$ in 2 mL of media per well. The following day the media was replaced with 2 mL of E6 media and 50% media changes were performed daily thereafter. Alternatively, to generate hPSC-derived micropatterned tissues, hPSC cultures at ~85% confluency were rinsed with PBS, dissociated with Accutase for 5 min at 37°C and collected via centrifugation at 1000 rpm for 5 min. Singularized hPSCs were then suspended in E8 media with 10 µM ROCK inhibitor and seeded onto micropatterned substrates at 100,000 cells/cm$^2$ in 2 mL of media per well. The following day the media was replaced with 2 mL of E6 media and 50% media changes were performed daily thereafter.

## Generation of micropatterned spinal cord neuroepithelial tissues

Caudalization of hPSC-derived cultures to generate micropatterend neuroepithelial tissues of the cervical/thoracic spinal cord was performed using a modified version of the deterministic *HOX* patterning protocol (*Lippmann et al., 2015*). Human PSC's were dissociated with Accutase and seeded onto matrigel-coated, 6-well TCPS plates at 150,000 cells/cm$^2$ in E8 media with 10 µM ROCK inhibitor. The following day the media was switched to E6 media for 24 hr, after which the media was supplemented with 200 ng/mL FGF8b (R and D Systems) to initiate differentiation into a NMP phenotype. Following 24 hr in E6 +FGF8 b media, the cells were dissociated with Accutase for 1:45 min and seeded onto TCPS plates at a 1:1.5 well ratio in E6 media with 200 ng/mL FGF8b, 3 µM CHIR (R and D Systems), and 10 µM ROCK inhibitor. The cells were allowed to remain in E6 +FGF8b + CHIR + ROCK inhibitor for 48 hr before the media was switched to E6 +FGF8b + CHIR. After 24 hr, the NMPs were dissociated with Accutase for 5 min and seeded onto micropatterned substrates at 150,000 cells/cm$^2$ or 250,000 cells/cm$^2$ in E6 media with 10 µM ROCK inhibitor. Six hours later, the media was supplemented with 1 µM Retinoic Acid (RA, R and D Systems) and 10 µM ROCK inhibitor to initiate transition to a neuroepithelial cell type. The next day the media was switched to E6 media with 1 µM RA and 50% media changes were performed every 24 hr for the next 2 days. Then, the media was supplemented with 1 µM RA, 2 µg/mL Sonic hedgehog, and 2 µM Purmorphomine (Shh, PM; R and D Systems) to induce ventralization. To further enhance ventralization, a 6 hr boost in Wnt signaling via media supplementation with 6 µM CHIR was applied either immediately before subculture or 72 hr after cell seeding onto micropatterned substrates but prior to RA +SHH + PM exposure. The neuroepithelial cells were maintained in E6 +RA + SHH + PM with 50% media changes daily.

## Neuroepithelial tissue image analysis algorithm and descriptor definitions

An automated workflow for quantitative analysis of micropatterned neuroepithelial tissue morphology was generated using MATLAB (MathWorks, *Supplementary file 2*). We first defined morphological characteristics of polarized neuroepithelial tissues. Neural rosettes in micropatterned tissues were identified by the presence of a coherent N-cadherin ring structure described in algorithm logic as halo (*Figure 2—figure supplement 1B*). Neural rosette morphology was further defined by the density of N-cadherin localization along constituent cell membranes with apico-basal polarity, described here as tendrils (*Figure 2—figure supplement 1C–D*). Areas of NECs that exhibited

punctate localization of N-cadherin absent the requisite ring structure and corresponding apico-basal polarity were classified as polarization foci (*Figure 1F*). Next, descriptors were established for variance in size and shape of zones of N-cadherin localization defined in the algorithm as measures of polarization area, fitted ellipse major axis length, and eccentricity. Further, a halo intensity ratio was applied to N-cadherin ring structures as a measure of the ratio of internal and external fluorescence intensity across the boundary of N-cadherin localization. Lastly, the apico-basal polarity of constituent cells surrounding apical N-cadherin localization was characterized as a measurement of cell membrane alignment angle compared to the tangent of the N-cadherin boundary (i.e. tendril angle incoherence 1 and 2). This measure was to be analyzed in conjunction with a measure of the tendril 1 and 2 density. Continuous variation in the above descriptors was accounted for to allow discontinuous response in classification of polarizing NECs as polarization foci or neural rosettes.

## Conducting automated analysis of neuroepithelial tissue images

Automated analysis of NEC polarization and neural rosette emergence was performed using 3D image stacks of single micropatterned neural tissues fluorescently stained for nuclei and N-cadherin. 3D image stacks were compiled from seven vertically segmented image acquisitions at 1.5 μm intervals using a 60X objective with $1024 \times 1024$ pixel resolution. Entire datasets were analyzed in batch fashion using the custom MATLAB program described above (*Figure 2C*; *Supplementary file 2*). First, images were pre-processed by applying a median filter to the maximum intensity projection. Next, regions of dense N-cadherin staining were identified as discrete blobs (i.e. polarization foci) using the process of maximally stable extremal regions (MSER) (*Matas et al., 2004*). After applying a smoothing filter, polarization foci area, eccentricity, and major axis of a fitted ellipse ('radius') was calculated for each region detected. Foci with area less than 176 μm$^2$ were excluded from further processing. To the mean image projection with a median filter, the location and intensity ratio of the N-cadherin halo was assessed. Identification of the halo served to both evaluate the coherence of the N-cadherin ring and establish a boundary for assaying arrangement of tendrils in relation to the halo. Once the halo boundary was established, the tendril density and tendril angle were measured by sampling 20 pixel squares at various points around the foci epicenter at a distance of 1.5X (*Thomson et al., 1998*) and 2X (*Eiraku et al., 2008*) the radius of the foci. Within each sampled area, pixel intensity was assayed to measure density of tendrils and a gradient orientation filter was used to calculate the directionality of tendrils relative to the halo boundary. For each foci identified, values for tendril angle 1 and 2 and tendril density 1 and 2 were calculated as averages from all points sampled. Collectively, these eight descriptors comprised an image descriptor vector.

## Developing a predictive, machine-learned classifier of polarization

The MATLAB Machine Learning Toolbox was used to train a classifier based on descriptor vectors from 70, randomly selected, 300 μm diameter (0.071 mm$^2$) circular tissues images (55). In addition to the algorithm analysis, these images were also examined independently by five NEC culture experts, who were asked to quantify the total number of N-cadherin$^+$ polarization areas (i.e. foci) and determine which should also be classified as rosettes. Agreement between 4 out 5 experts was required to designate any polarization foci or rosette as a 'ground truth'. Then, the human ground truth and the algorithm's descriptor data for each identified N-cadherin polarization area in the 70 tissues (235 total identified areas) was used to learn a classifier using logic regression (*Supplementary file 3*). Comparison of each human expert's analyses versus the consensus ground truth revealed a human polarization foci and rosette identification error rate of 21.19 ± 3.35% and 13.05 ± 2.74%, respectively (Table S1). In comparison, the classification function exhibited lower error rates of 14.35% and 8.86%, respectively.

To further test the classifier, a second round of human expert and algorithm analyses was conducted on an additional 35 neuroepithelial tissues selected randomly and distributed evenly across all morphologies. The human experts and automated image analysis algorithm identified 148 N-cadherin polarization areas total. The human versus classifier polarization and rosette error rates were 19.05 ± 14.61% vs. 20.27% and 9.93 ± 6.96% vs. 19.56%, respectively (Table S2). This suggest that the classifier generalizes well to new morphologies although the classifier's rosette identification error rate increased potentially due to the variance in tissue morphologies. Still, the image analysis

algorithm and classifier workflow were deemed acceptable for detecting trends in singular rosette emergence given the magnitude of differences observed in *Figure 2B*.

## Clickable peptide bioconjugate synthesis

Bioconjugate ligands were synthesized to undergo copper-free 'click' reactions with azide-functionalized PEG-grafted micropatterned substrates in accordance with our published protocol (*Knight et al., 2015*). Fluorescein isothiocyanate (FITC)-tagged and a non-fluorescent version of the FGF-2 heparin-binding peptide (*Hudalla and Murphy, 2010*) were purchased from Genescript (Piscataway, NJ). Peptides were designed with a terminal cysteine residue to enable Michael-type addition of a clickable heterobiofunctional linker molecule, consisting of dibenzocyclooctyne (DBCO) with a maleimide-PEG4 crosslinker (DBCO-PEG$_4$-Maleimide, Click Chemistry Tools). Peptides were added to a solution of Tris(2-carboxethyl)phosphine hydrochloride (TCEP) in sterile water pH 7.0 for 1 hr at room temperature. DBCO-PEG$_4$-Maleimide in DMF was then added to the solution in 4:1 molar excess and allowed to react while shaking for 1 hr at room temperature. Upon completion, the reaction was pipetted onto the gel bed of a size-exclusion chromatography (SEC) column packed with Bio-Gel P-2 Gel (Bio-Rad) and allowed to enter the column under gravity flow. The reaction solution was passed through the column using 100 mL sterile water and eluted into one hundred, 1 mL aliquots. Eluted aliquots were concentrated via freeze-drying and analyzed with UV-Vis spectrophotometry for peaks at 309 and 495 nm wavelengths, corresponding to the presence of DBCO and FITC respectively, as well as by a Micro-BCA Assay (Promega) (*Figure 6—figure supplement 2*). Peptide bioconjugate containing aliquots were stored lyophilized at −20°C, and upon reconstitution in PBS, the bioconjugate concentration was re-assessed by comparing the UV-Vis DBCO measurement to a standard curve of known concentrations.

## Azide functionalization of grafted poly(ethylene glycol) Brushes

PEG-MA-grafted micropatterned arrays underwent Steglich esterification reactions to substitute side-chain hydroxyl groups with bromine, as previously published(*Knight et al., 2015*). Freshly washed substrates were first transferred to Schlenk flasks into which a degassed solution of N,N-Dimethylformamide (Sigma Aldrich) containing 1-ethyl-3-(3-dimethylaminopropyl)carbodiimide hydrocholoride (EDC) (Thermo Fisher) and bromoacetic acid (Sigma Aldrich) was transferred. Esterification was initiated by injection of 4-(Dimethylamino)pyridine (DMAP) (Sigma Aldrich) in DMF and was allowed to continue for 1 hr at room temperature. The reaction was terminated by removal of the substrates and neutralization of the reaction solution. Modified substrates were rinsed separately in water and ethanol, and then dried under inert gas. Once transferred to a scintillation vial, substrates underwent nucleophilic substitution of both terminal and side-chain bromine groups with azido groups via addition of DMF solution with sodium azide (Sigma Aldrich) for 16 hr at 37°C. Termination of the reaction was achieved by removal of the substrates and neutralization of the reaction solution.

## In Situ modification of micropatterned neural tissues arrays

After seeding hPSCs to generate micropatterned neuroepithelial tissue arrays as previously described, E6 media supplemented with 10 µM HBP-DBCO bioconjugate was added on day 3 or 4. After 24 hr, the media was replaced with 2 mL E6 media and 50% media changes were performed daily thereafter.

## Gene expression analysis of micropatterned neuroepithelial tissues' Regional Phenotype

Total messenger RNA was isolated from caudalization day 2 cells (*Figure 5D*) in well plates and microarrayed forebrain (Day 4, *Figure 5A*) and spinal (Day 10, *Figure 5D*) neuroepithelial tissues using the PureLink RNA mini kit (Invitrogen). For qRT-PCR, cDNA synthesis was conducted using the SuperScript IV First-Strand Synthesis System (Thermo Fisher), and gene expression was quantified using TaqMan Gene Expression Assays on a BioRad CFX96 detection unit (Table S4). Data was presented as $\Delta\Delta C_t$ values relative to *RSP18* and normalized to caudalization day 2 cells (*Figure 4—figure supplement 1A*).

## Quantification of cell density

Quantification of cell density in micropatterned neuroepithelial tissues was performed using image segmentation packages in CellProfiler (BROAD Institute) to identify and count DAPI stained nuclei in 3D image stacks. Image segmentation was also used to identify and quantify the number of nuclei stained positively for transcription factors Pax6, Otx2, Nkx6.1, and Olig2. The number of nuclei expressing the transcription factor of interest was compared to the total DAPI nuclei present to estimate percentages of regionally patterned NECs.

## Immunocytochemistry and microscopy

Micropatterned tissues were rinsed with PBS and fixed with 4% paraformaldehyde (PFA, Sigma Aldrich) in PBS for 10 min at room temperature. Following three rinses with PBS, the micropatterned substrates were blocked and permeabilized in PBS with 5% Donkey Serum and 0.3% Trition X-100 (TX100; Thermo Fisher) (PBS-DT) for 1 hr at room temperature. Next, the tissues were incubated with primary antibodies diluted in PBS-DT for 2 hr at room temperature or overnight at 4°C. Then, the substrates were rinsed in PBD-DT three times for 15 min each. Secondary antibodies were diluted in PBS-DT and exposed to the tissues for 1 hr at room temperature. A list of primary antibodies and dilutions can be found in Table S3. Tissue nuclei were subsequently stained with 300 nM 4′,6-diamidino-2-pheny-lindoldihydrochloride (DAPI, Sigma Aldrich) for 30 min followed by three 15 min PBS washes. The micropatterned substrates were mounted on glass coverslips using Prolong Gold Antifade Reagent (Thermo Fisher). Brightfield images of micropatterned tissues were obtained using a Nikon TS100 microscope with a ME600L camera. Fluorescence images were obtained using a Nikon A1R confocal microscope.

## Supplementary materials

Supplementary materials include: supplemental figures for *Figures 1*, *2*, *4*, *5* and *6*; Table S1 and S2 of human vs. algorithm error rates using the automated neuroepithelial image analysis algorithm and machine learning classifier (*Supplementary file 1*); tables of antibodies and gene expression assays (*Supplementary file 1*); image analysis algorithm MATLAB code (*Supplementary file 2*); machine learning classifier MATLAB workspace (*Supplementary file 3*); source data for quantitation in all Figures/supplemental figures (*Figure 1—source data 1*; *Figure 2—source data 1*; *Figure 3—source data 1*; *Figure 4—source data 1*; *Figure 5—source data 1*; *Figure 6—source data 1*).

# Acknowledgements

This publication was developed, in part, with STAR center grant 83573707 from the U.S. Environmental Protection Agency (EPA) but has not been formally reviewed by the EPA. The views expressed in this document are solely those of RSA and colleagues and do not necessarily reflect those of the Agency, and the EPA does not endorse any products or commercial services mentioned in this publication. Authors would also like to thank Sanika Gawhane for assistance with preliminary development of the image analysis algorithm.

# Additional information

### Funding

| Funder | Grant reference number | Author |
| --- | --- | --- |
| National Science Foundation | CCF-1418976 | Rebecca M Willett |
| National Institutes of Health | 1 U54 AI117924-01 | Rebecca M Willett |
| National Science Foundation | IIS-1447449 | Rebecca M Willett |
| Environmental Protection Agency | 83573701 | Randolph Scott Ashton |
| National Institute of Neurological Disorders and Stroke | R21NS082618 | Randolph Scott Ashton |
| Burroughs Wellcome Fund | 1014150 | Randolph Scott Ashton |

| National Science Foundation | 1651645 | Randolph Scott Ashton |
| National Institute of Neurological Disorders and Stroke | R33NS082618 | Randolph Scott Ashton |

The funders had no role in study design, data collection and interpretation, or the decision to submit the work for publication.

## Author contributions

Gavin T Knight, Conceptualization, Data curation, Formal analysis, Validation, Investigation, Visualization, Methodology, Writing—original draft, Writing—review and editing, Helped conceived the study, Performed substrate synthesis and cell-culture experiments, Helped design the parameters for automated neural tissue analysis, Contributed to manuscript preparation; Brady F Lundin, Data curation, Investigation, Visualization, Methodology, Assisted with substrate synthesis and cell-culture experiments, Contributed to manuscript preparation; Nisha Iyer, Data curation, Formal analysis, Investigation, Helped performed cell-culture experiments, Conducted gene expression analysis; Lydia MT Ashton, Data curation, Software, Formal analysis, Methodology, Writing—review and editing, Developed the machine learning classifier, Contributed to manuscript preparation; William A Sethares, Formal analysis, Methodology, Helped design the parameters for automated neural tissue analysis, Helped program the image analysis algorithm, Contributed to manuscript preparation; Rebecca M Willett, Data curation, Formal analysis, Methodology, Helped design the parameters for automated neural tissue analysis and program the algorithm, Contributed to manuscript preparation; Randolph Scott Ashton, Conceptualization, Data curation, Supervision, Funding acquisition, Investigation, Visualization, Methodology, Writing—original draft, Project administration, Writing—review and editing, Conceived the study, Helped design all experiments and the parameters for automated neural tissue analysis, Contributed to manuscript preparation

## Author ORCIDs

Gavin T Knight ⓘ https://orcid.org/0000-0003-4633-1924
Brady F Lundin ⓘ https://orcid.org/0000-0003-0708-8140
Rebecca M Willett ⓘ http://orcid.org/0000-0002-8109-7582
Randolph Scott Ashton ⓘ http://orcid.org/0000-0002-6842-7022

## Decision letter and Author response

Decision letter https://doi.org/10.7554/eLife.37549.029
Author response https://doi.org/10.7554/eLife.37549.030

# Additional files

## Supplementary files

• Supplementary file 1. Image analysis data and reagent tables. Table S1 and S2 provide summary statistics on human vs. algorithm (i.e. the automated neuroepithelial image analysis algorithm and machine learning classifier) error rates. Table S1 reports analysis conducted on 300 μm diameter (0.071 mm$^2$) circular tissue images, and Table S2 reports analysis conducted on 35 neuroepithelial tissue images selected randomly and distributed across all morphologies. Table S3 and S4 provide catalog and usage information for primary antibodies and QPCR primer sets used in this study.
DOI: https://doi.org/10.7554/eLife.37549.024

• Supplementary file 2. Image analysis algorithm. Folder provides instructions, that is 'readme.txt', and code files for batch analysis of N-cadherin immunostained neuroepithelial tissue images using MATLAB. This folder can be used to create image descriptor vectors.
DOI: https://doi.org/10.7554/eLife.37549.025

• Supplementary file 3. Machine learning classifier workspace. Folder provides MATLAB workspace files that can be uploaded to apply our custom classifier to image descriptor vector data sets. This is used to automatically classify whether N-cadherin$^+$ foci within a neuroepithelial tissue image are polarization foci or rosettes.
DOI: https://doi.org/10.7554/eLife.37549.026

• Transparent reporting form
DOI: https://doi.org/10.7554/eLife.37549.027

All data generated or analysed during this study are included in the manuscript and supporting files. Source data files have been provided for Figures 1 to 6.

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
