## [Decision Letter]

Thank you for submitting your article "Engineering Induction of Singular Neural Rosette Emergence within hPSC-derived Tissues" for consideration by *eLife*. Your article has been reviewed by two peer reviewers, and the evaluation has been overseen by Didier Stainier as the Senior Editor and Reviewing Editor. The following individuals involved in review of your submission have agreed to reveal their identity: Alfonso Martinez Arias (Reviewer #1).

In this manuscript, the authors help introduce rigor and standards to neural organoids and especially to those modeling early stages of neurogenesis. The work is novel, interesting and well done.

Usually, we provide a condensed summary of the reviews but we believe that, in this case, you would benefit from receiving the full reviews verbatim, which are attached below.

*Reviewer #1:*

The field of neural 'organoid' is, for the most part, one of much noise and little clarity about what is actually going on in the cultures so it is very good to see a study like this one in which the authors apply rigour and high standards to the reconstruction of the early stages of neurogenesis in vitro. In particular one of the problems that bedevils the field is that of reproducibility and it is in tackling it that the authors make some important findings. Specifically the different biophysical parameters that are required for brain and spinal cord. This is a significant and important contribution to the efforts of creating ex vivo models for the development of the nervous system that will help advances in disease and drug screening.

I would have liked to see if the rosettes express FoxA2 and SHH (floor plate markers).

*Reviewer #2:*

In this work, the authors combined geometric confinement and E6 based induction protocol to generate singular neural rosette tissue using hESCs. The method reported in this work is very novel and potentially useful for future mechanistic investigations; however the main text is somewhat disorganized, making it difficult to read and understand transitions between figures. Below are my main concerns and questions.

1) There is no mechanistic study (such as using genetics approach) in this study. It still remains largely unclear how hESCs sense the circular geometry/confinement and become polarized during the rosette formation process. Why are the cells sensitive to the circular pattern size during the induction process? These are important questions that need to be properly investigated.

2) More detailed quantification and immunostaining/live cell imaging need to be performed to better understand the neural rosette formation process. In the current manuscript, only Pax6 and N-cadherin are used to label neural rosette. More cell lineage/fate markers and more importantly, cell polarity proteins need to be quantified.

3) There are some overstatements in this study. For example, in Figure 6G, the authors stated that there is lumen formation in the rosette. But the black cavity doesn't necessarily look like a real lumen to me. Again, some staining for apical proteins such as aPKC and Par6 would be very helpful. In the subsection “Neural rosette emergence is regulated by tissue morphology”, the definition of '+1 rosette' should be carefully examined, because the clear definition for polarization is lacking. Again, more polarity related proteins should be studied in this work.

4) The authors should confirm their neural rosette protocol using hiPSC lines.

5) In this work, the neural rosette formation is sensitive to the mechanical cue (the geometry and size of the pattern). The connection between this in vitro system and in vivo neural rosette formation should be better elucidated. Is there any evidence showing the neurulation process in vivo is affected by mechanical cues?

6) How does the pattern size and shape affect the neural rosette formation in this work? Does the pattern size affect certain signaling pathway (BMP? TGF-β?) and then affect the downstream neural activation? Or does the mechanical cue affect the certain transcription factors (YAP/TAZ) directly? More mechanistic studies about the mechanism will certainly strengthen this current study.

---

## [Author Response]

Reviewer #1:[…] I would have liked to see if the rossettes express FoxA2 and SHH (floor plate markers).

We have immunostained for FoxA2 and SHH to determine if a dorsoventral axis formed spontaneously within the arrayed rosettes. This was motivated by observation of such an occurrence within suspended neuroepithelial cysts formed from mouse embryonic stem cells (Meinhardt et al., 2014). However, we did not observe FoxA2 and SHH immunostaining under our current culture conditions. We hypothesize that a higher concentration of SHH would be needed to induce expression of floor plate markers within our arrayed rosettes (Pusapati et al., 2018).

Reviewer #2:In this work, the authors combined geometric confinement and E6 based induction protocol to generate singular neural rosette tissue using hESCs. The method reported in this work is very novel and potentially useful for future mechanistic investigations; however the main text is somewhat disorganized, making it difficult to read and understand transitions between figures. Below are my main concerns and questions.

We apologize for the reviewer’s difficulty reading the manuscript. We acknowledge that is a lengthy manuscript but see this as unavoidable given the interdisciplinary nature of the science. We have tried to make the Results and Discussion sections as straight-forward as possible. We have made revisions throughout to simplify sentence structure and improve clarity.

1) There is no mechanistic study (such as using genetics approach) in this study. It still remains largely unclear how hESCs sense the circular geometry/confinement and become polarized during the rosette formation process. Why are the cells sensitive to the circular pattern size during the induction process? These are important questions that need to be properly investigated.

Under our differentiation protocols, which have been published previously, human pluripotent stem cells uniformly differentiate into neuroepithelial cells (Lippmann, Estevez-Silva and Ashton, 2014; Lippmann et al., 2015). Polarization of neuroepithelial cells is a characteristic of their phenotype, and there are numerous papers that provide a mechanistic understanding of this phenomenon (Nishimura, Honda and Takeichi, 2012; Copp, Greene and Murdoch, 2003; Vladar, Antic and Axelrod, 2009). As shown in Figure 5B, we observed instances of single rosette formation on all pattern sizes; however, the propensity for generating only a single rosette increases significantly in the identified micropattern diameter ranges. We explicitly state, “Hence, the micropatterned tissue morphology does not instruct rosette morphogenesis, but restricts the tissue area to statistically favor emergence of a single rosette in accordance with innate NEC properties”.

Thus, there is no mechanistic or genetic study to conduct since the geometric confinement appears to be conditionally favorable and not instructive.

2) More detailed quantification and immunostaining/live cell imaging need to be performed to better understand the neural rosette formation process. In the current manuscript, only Pax6 and N-cadherin are used to label neural rosette. More cell lineage/fate markers and more importantly, cell polarity proteins need to be quantified.

The purpose of this manuscript is not to “understand the neural rosette formation process”, but to develop a novel engineering approach to control its emergence. Numerous papers including our own have concluded that Sox2/Pax6 positive cells with polarized-localization of N-cadherin in a rosette structure are definitive neuroepithelial/neural rosette makers (Lancaster et al., 2013; Lippmann, Estevez-Silva and Ashton, 2014; Lippmann et al., 2015; Shi et al., 2012). Furthermore, we show basal deposition of Laminin proteins indicative of their ability to generate a basement membrane (Figure 5). We do not believe that additional proof is necessary, but we have also provided immunostaining to show apical localization of ZO-1, cortical F-actin, and mitotic cell division. The following text was inserted:

“Co-localization of F-actin, ZO-1 tight junction proteins, and phospho-H3^+^ mitotic nuclei within these polarized structures further supports the use of polarized N-cadherin as a surrogate marker of neural rosette emergence (Figure 1—figure supplement 1A).”

3) There are some overstatements in this study. For example, in Figure 6G, the authors stated that there is lumen formation in the rosette. But the black cavity doesn't necessarily look like a real lumen to me. Again, some staining for apical proteins such as aPKC and Par6 would be very helpful.

As mentioned in the previous response, we and others have used N-cadherin as a marker of neural rosette formation in Sox2/Pax6 cultures. Figure 6Gi is an XY cross-section through the middle of the hemispherical cell cluster, and Figure 6Giii is a XZ cross-section of the same cell cluster. The elongate Pax6^+^ nuclei in Figure 6Gi clearly show polarization to a central region that is confirmed as lumen in 3D by the void space in Figure 6Giii. To further corroborate lumen formation, we have added a 3D animated deconstruction of the immunostained cell cluster. This has been noted in the text (Discussion, last paragraph), Figure 6’s legend, and added as Figure 6—figure supplement 5.

In the subsection “Neural rosette emergence is regulated by tissue morphology”, the definition of '+1 rosette' should be carefully examined, because the clear definition for polarization is lacking. Again, more polarity related proteins should be studied in this work.

Figure 1F provides illustrations of what we interpret as a ‘polarization foci’ and a ‘neural rosette’. In the Materials and methods and Supplementary files, we detail quantitative image analysis parameters that define a rosette versus a polarization when developing our image analysis algorithm. As stated, micropatterned cell clusters were identified as ‘+1 rosette’ when they contained “≥ 1 rosette with ≥ 1 additional polarization foci”. We do not know how to be more explicit. As indicated in our response to reviewer #2’s point 1, we have shown additional immunostaining (e.g. ZO-1, cortical F-actin, marker of cell division) to validate that polarized N-cadherin is a proper indicator of neural rosette emergence.

4) The authors should confirm their neural rosette protocol using hiPSC lines.

We have conducted this experiment, and added the following text:

“Interestingly, while these results were consistent for tested hESC lines, micropatterned E6 differentiation of D90A *SOD1* mutant ALS-iPSCs also showed optimal singular, forebrain rosette emergence within 250μm diameter circular tissues but at a lower (26%) efficiency (Figure 5—figure supplement 2). This corresponded to many instances where tissues displayed a non-uniform Pax6^+^ NEC phenotype, indicating that line-to-line variations in neural induction efficiency must be overcome for effective micropatterning of singular neural rosette emergence.”

5) In this work, the neural rosette formation is sensitive to the mechanical cue (the geometry and size of the pattern). The connection between this in vitro system and in vivo neural rosette formation should be better elucidated. Is there any evidence showing the neurulation process in vivo is affected by mechanical cues?

As mentioned in the first paragraph of subsection “Neural rosette emergence is regulated by tissue morphology”, “Biophysical studies have inextricably linked developmental morphogenesis events such as neural tube formation with tissue biomechanics (Davidson and Keller, 1999; Nishimura, Honda and Takeichi, 2012)”. Nishimura, Honda and Takeichi (2012) uses chick embryos to investigate planar-cell-polarity pathway activity during neural tube closure. This provides a mechanistic link between neuroepithelial cell polarization/neural tube closure/in vivo neural rosette formation and activation of Rho kinase leading to actomyosin contraction (Figure 7C in Nishimura, Honda and Takeichi, 2012). This is the biomechanical link to neural rosette formation. We would like to reiterate that we do not claim that the geometry or size of the pattern instructs neural rosette formation (stated explicitly in the subsection “NEC biomechanical properties vary by regional phenotype and affect rosette emergence behavior”). Instead, we show that it provides favorable conditions for emergence of a single rosette based on intrinsic properties of the micropatterned neuroepithelial cells.

6) How does the pattern size and shape affect the neural rosette formation in this work? Does the pattern size affect certain signaling pathway (BMP? TGF-β?) and then affect the downstream neural activation? Or does the mechanical cue affect the certain transcription factors (YAP/TAZ) directly? More mechanistic studies about the mechanism will certainly strengthen this current study.

See response to reviewer #2, points 2 and 5.

References:

A.J. Copp, N.D.E. Greene, J.N. Murdoch, The genetic basis of mammalian neurulation, Nat Rev Genet. 4 (2003) 784–793.

G.V. Pusapati, J.H. Kong, B.B. Patel, M. Gouti, A. Sagner, R. Sircar, et al., G proteincoupled receptors control the sensitivity of cells to the morphogen Sonic Hedgehog, Sci Signal. 11 (2018) eaao5749

] E.K. Vladar, D. Antic, J.D. Axelrod, Planar cell polarity signaling: the developing cell's compass, Cold Spring Harbor Perspectives in Biology. 1 (2009) a002964–a002964.